# Development of a High Internal Phase Emulsion of Antarctic Krill Oil Diluted by Soybean Oil Using Casein as a Co-Emulsifier

**DOI:** 10.3390/foods10050917

**Published:** 2021-04-22

**Authors:** Yunhang Liu, Dongwen Fu, Anqi Bi, Siqi Wang, Xiang Li, Xianbing Xu, Liang Song

**Affiliations:** 1School of Food Science and Technology, Dalian Polytechnic University, Dalian 116034, China; lyhliuyunhang@163.com (Y.L.); fdw2yy@163.com (D.F.); bianqi96@163.com (A.B.); 18340854368@163.com (S.W.); lx910702@163.com (X.L.); xianbingxu@dlpu.edu.cn (X.X.); 2National Engineering Research Center of Seafood, Dalian 116034, China

**Keywords:** high internal phase emulsions, Antarctic krill oil, phospholipids, casein, co-emulsifier

## Abstract

Antarctic krill oil (AKO) with 5–30% (*w/w*) dilution by soybean oil was co-emulsified by phospholipids (PLs) naturally present in AKO and 2% (*w/w*) casein in the aqueous phase to prepare high internal phase emulsions (HIPEs). The results showed that raising the AKO level resulted in concave-up changes in the mean size of oil droplets which became more densely packed. Confocal laser scanning microscopy (CLSM) and cryo-scanning electron microscopy (cryo-SEM) micrographs revealed that PLs at higher concentrations expelled more casein particles from the oil droplet surface, which facilitated the formation of a crosslinked network structure of HIPEs, leading to reduced mobility of water molecules, extended physical stability, and somewhat solid-like behavior. The rheological analysis showed at lower levels of AKO promoted fluidity of emulsions, while at higher levels it increased elasticity. Lastly, increasing the AKO level slowed down the oxidation of HIPEs. These findings provide useful insights for developing HIPEs of highly viscous AKO and its application in foods.

## 1. Introduction

Antarctic krill oil (AKO) contains considerable amounts of bioactive minor components such as astaxanthin, sterols, tocopherols, vitamin A, flavonoids, and minerals [1], as well as a high content of Eicosapentaenoic Acid (EPA) and Docosahexaenoic Acid (DHA) that are present in phospholipids (PLs, 39.89% to 80.69%), which mainly comprise phosphatidylcholine (PC, 44.58% to 99.80% of total PLs) and phosphatidylethanolamine (PE, 0.20% to 24.74% of total PLs) [2]. Various delivery systems for AKO were developed to improve its stability and accessibility, including microemulsions [1], liposomes [3], microcapsule [4], nanostructured lipid carriers [5], yeast cells [6], among others. However, these delivery systems generally have a low loading capacity of AKO. Therefore, there is a need to develop a new delivery system with high loading capacity for AKO.

High internal phase emulsions (HIPEs) refer to emulsions that have a large volume fraction of dispersed phase, which is greater than 0.74 [7]. Due to their high oil content, oil-in-water (O/W) HIPEs (hereinafter HIPEs) can carry high concentrations of hydrophobic bioactives, such as curcumin [8], lutein [9], *β*-carotene [7]. In addition, the semi-solid viscoelastic structure of HIPEs that is formed by high accumulation and crosslinking of the oil phase makes the emulsions highly stable and resistant to creaming. Because of these characteristics, HIPEs have been used to carry a high content of oil-soluble active substances [7] and convert liquid oil into semi-solid term without changing the chemical structure for applications in spreads [10], and used as food-grade 3D printing materials [11].

The stability and rheological properties of HIPEs are affected by experimental conditions, such as temperature, pH, ionic strength, and additives [12]. It has long been known that surfactants, which are solubilized in the continuous phase, can stabilize HIPEs by slowing down or preventing droplet coalescence. For example, Williams [13] studied effects of 8 surfactants and 22 cosurfactants on the stability of a styrene-based HIPE. Because more and more well-informed consumers are seeking organic or clean label products [14,15], particles of biopolymers such as proteins, polysaccharides, and lipids have been suggested as an alternative to synthetic surfactants traditionally used in highly concentrated edible emulsions [16,17,18]. To enhance the capability of biopolymer particles to stabilize HIPEs, the particles usually need to be physically or chemically modified [18,19,20]. At present, preparation of highly effective biopolymer particle stabilizers is still facing challenges. For example, the organic reagents introduced in chemical modification processes are difficult to remove, which may cause potential harm to human health. Besides, it is difficult to produce on the commercial scale nanoparticles and microparticles that are both effective and acceptable for use in food products [21].

Previous studies [22,23] found that the combination of natural amphiphilic phospholipids and proteins exhibited much stronger emulsifying properties than each individual emulsifier. Chen et al. [24] reported that gliadin particles modified with phospholipids could improve the foaming property of the particles, and the interfacial elasticity and quercetin encapsulation efficiency of emulsions prepared using the particles increased with increasing the phospholipid content. Xue and Zhong [25] found the blend of sodium caseinate and lecithin resulted in droplets of thyme oil nanoemulsions that were significantly smaller and more narrowly distributed than those prepared using sodium caseinate or lecithin alone. The strengthened capability of combined phospholipids and proteins to form stable food colloids was postulated as the interactions between phospholipids and proteins attributed mainly to hydrogen bonds and hydrophobic forces [22].

Based on the abovementioned observations, we hypothesized that PLs naturally present in AKO may interact with an added protein in the oil–water interface and thus enhance the protein’s emulsifying properties. On this basis, in the present study, food-grade HIPEs of AKO carried by soybean oil were prepared using casein as a co-emulsifier together with PLs in AKO. To understand a possible underlying mechanism of the emulsion formation, the microstructure, particle size, rheological properties, and stability of the emulsions were investigated.

## 2. Materials and Methods

### 2.1. Materials

Casein was obtained from Aladdin (Shanghai, China). Antarctic krill oil (AKO) was obtained from Kang Jing Marine Biotechnology (Shandong, China). Soybean oil was obtained in a local supermarket (Dalian, China). Nile Red was purchased from BBI Life Sciences (Shanghai, China) and Nile Blue A from Aladdin (Shanghai, China). All other chemicals used were analytical reagents.

### 2.2. Preparation of HIPEs

The HIPEs were prepared according to a previous study with slight modifications [26]. A 4.0% (*w/w*) casein suspension was stirred in distilled water overnight and its pH was then adjusted to 6.0 with 0.1 M NaOH or HCl solution. The obtained casein suspension was diluted to 2, 1, 0.5, and 0.25% (*w/w*) by distilled water at pH 6.0. Then, 30 mL of soybean oil were mixed with 10 mL of each casein suspension to obtain a 3:1 volume ratio (φ = 75%) and sheared at 8000 rpm for 60 s using a T25 digital Ultra Turrax (IKA, Staufen, Germany) mixer. Similarly, 5, 10, 15, 20, 25, and 30% (*w/w*) of soybean oil was replaced by AKO. The obtained HIPEs (Table 1) were stored at 4 °C for further analysis. The phospholipid content in the oil phase was determined by molybdenum blue colorimetry according to Wu et al. [27]. The EPA and DHA content in the oil phase was determined by gas chromatography (GC) according to Yin et al. [28]. The viscosity of mixed oil and bulk AKO were measured at 25 °C using a DV-1 digital rotating viscometer with a CPA-40Z stainless plate (AMETEK Brookfield, Middleborough, MA, USA).

### 2.3. Observation of Microstructure of HIPEs

#### 2.3.1. Optical Microscope

Inverted light microscope was applied to observe the structure of HIPEs and oil droplets’ distribution according to Chen et al. [29]. The samples were placed in glass slides, covered with cover glass and compacted gently, and the microstructure was then observed by a Revolve Hybrid Microscope (Echo, San Diego, CA, USA) and recorded using a retina screen iPad Pro tablet (Apple Inc., Cupertino, CA, USA) attached to the microscope. The magnification was 20 times.

#### 2.3.2. Confocal Laser Scanning Microscopy (CLSM)

A SP8 confocal laser scanning microscope (LEICA, Wetzlar, Germany) was used to characterize the microstructure of the HIPEs according to a previous study with slight modifications [30]. The HIPEs were dyed with a mixed fluorescent dye solution consisting of 1 mg/mL Nile Red (for lipid stain) and 1 mg/mL Nile blue A (for protein particles stain). The stained HIPEs were then observed by a 63× oil immersion lens, with an argon krypton laser having an excitation of 488 nm (for Nile Red) and a helium neon laser (He/Ne) with excitation at 633 nm (for Nile blue A).

#### 2.3.3. Cryo-Scanning Electron Microscopy (Cryo-SEM)

The surface morphology of HIPEs was observed by using a cryo-scanning electron microscopy (Hitachi, Chiyoda, Tokyo, Japan) with an accelerating voltage of 10 kV. The HIPEs were put on the sample stage and frozen in liquid nitrogen (−196 °C). The frozen HIPE samples were cut with a cooled knife in a freezing separation chamber and sublimed at −80 °C for 40 min. After sublimation, the samples were sputter-coated with a gold-palladium alloy at 10 mA for 60 s and transferred to a cryo-SEM to observe the structures.

### 2.4. The Size Distribution of Oil Droplets

The size distribution of oil droplets in freshly prepared samples was determined using static light scattering (Mastersizer 3000, Malvern Instruments, Worcestershire, UK). Particle size data was reported as the volume-weighted mean diameter (d_43_).

### 2.5. Measurement of Rheological Properties

Rheological properties of HIPEs were determined by using a Discovery HR-1 rheometer (TA Instruments, New Castle, DE, USA) with aluminum Peliter plates (998332, TA Instruments, New Castle, DE, USA). The HIPE samples were loaded into the rheometer and remained unperturbed for 2 min before tests. Amplitude sweeps were performed at a fixed frequency of 1 Hz to determine a linear viscoelastic region (LVR). The frequency sweeps were performed between 0.1 and 100 Hz at a fixed 1.00% strain in the LVR to measure the elastic and viscous modulus (G′ and G″) of HIPEs. The measurements were carried out at 25 °C.

### 2.6. Low-Field Nuclear Magnetic Resonance (LF-NMR)

The LF-NMR measurement of HIPEs were performed using the LF-NMR instrument (Suzhou Niumag Analytical Instrument Corporation, Suzhou, China) with all samples balanced at 25 °C for 30 min. Transversal relaxation (*T*_2_) was measured by the Carr−Purcell−Meiboom−Gill (CPMG) pulse sequence with a time delay between 90° and 180° pulses (τ-value) of 300 μs. Data from 1000 echoes were acquired from 8 scan repetitions. The repetition time between two succeeding scans was set to 3000 ms. Single-exponential fittings of CPMG decay curves were performed by using MultiExp Inv analysis software (Suzhou Niumag Analytical Instrument Co., Suzhou, China). Single-exponential fitting analysis was performed on the relaxation data in the software of simultaneous iterative reconstruction technique (SIRT) algorithm.

### 2.7. Measurement of Stability

#### 2.7.1. Centrifugation Stability

The centrifugation stability of HIPEs was investigated according to a method reported previously [31]. Briefly, 5 g of HIPEs were added to a centrifugal tube and centrifuged at 3050 g for 30 min. Emulsions were separated into a creamed layer at the top and a transparent serum layer at the bottom. The extent of creaming was characterized by the creaming index (CI, %) expressed by Equation (1) [32].
(1)$$\mathrm{CI}\left(\%\right)=\frac{\mathrm{Height}\;\mathrm{of}\;\mathrm{serum}}{\mathrm{Total}\;\mathrm{height}\;\mathrm{of}\;\mathrm{emulsion}\;}\times 100$$

#### 2.7.2. Storage Stability

The HIPEs were sealed in transparent glass bottles and incubated at 4 °C over 30 d to monitor their storage stability. The appearance photos of the HIPEs were taken and the microstructure of the HIPEs were observed by cryo-SEM through the method in Section 2.4 after 30 d.

#### 2.7.3. Oxidative Stability

The oxidation induction time of the HIPEs was measured by Rancimat (892, Metrohm, Herisau, Switzerland) according to literatures [33,34]. In brief, a weighted amount of an HIPE (3.0 ± 0.1 g) was added into each reaction tube and subjected to accelerated oxidation at 120 °C under a 20 L/h air flow that passed through the sample and then through a measuring vessel containing 50 mL of deionized water. The increase in conductivity of the water was measured as a function of time. The oxidation induction time which reflected the oxidation stability of the emulsion was then obtained from the conductivity curve.

### 2.8. Statistics

All experiments were performed in independent three replicates, and measurements were performed in triplicate. Statistical analyses were performed by using an analysis of variance (ANOVA) procedure of the SPSS 19.0 statistical analysis program (Armonk, NY, USA), and the differences between means of the trials were detected by a least significant difference (LSD) test (*p* < 0.05).

## 3. Results

### 3.1. Formation and Microstructure of HIPEs of Mixed Soybean oil and AKO

#### 3.1.1. Determination of the Casein Concentration in the Aqueous Phase

The effect of the casein concentration in the continuous (aqueous) phase on the formation of HIPEs was investigated by using soybean oil as the oil phase. Appearance and optical micrographs of high internal phase emulsions (HIPEs) (φ = 0.75) of soybean oil at different casein concentrations (0.25–4%, *w/w*) in the continuous phase are shown in Figure 1A. According to this figure, HIPEs could be successfully prepared when the casein concentration was greater than 0.5%, and demulsification was observed when it was 0.25%. With the increase in the casein concentration, the size of the oil droplets gradually decreased and the resultant emulsions tended to be more homogeneous and stable (Figure 1B). When the casein concentration was greater than 2%, the size of oil droplets did not change obviously, indicating that casein particles were sufficient to cover the oil droplets. Therefore, the 2% (*w/w*) casein suspension was chosen in the following experiments. A similar finding was reported for HIPEs stabilized by wheat gluten [10] and whey protein and gliadin particles [35]. This dependence was attributed to the fact that, at low protein concentrations, there were insufficient protein particles to completely cover the surface of oil droplets formed during homogenization, leading to coalescence of some oil droplets [36].

#### 3.1.2. Effects of the AKO Level on Formation and Microstructure of the HIPEs

Figure 2 illustrates appearance (A), CLSM observation (B), cryo-SEM observation (C), and volume mean particle size (D) of HIPEs with different levels of AKO in the oil (soybean oil) phase. The abundant PLs in AKO leads to its high viscosity, thence soybean oil is used as a diluent of AKO, because it is cheap and has good oxidation stability and low viscosity. The corresponding PLs content and viscosity in each mixed oil is listed in Table 2. Figure 2A shows that the color of HIPEs gradually deepened to orange red as the AKO level increased, which resulted from the dark red color of AKO. Visual observation also suggested that the HIPEs displayed as smooth cream-like sols at 0–10% levels of AKO, whereas they appeared gel-like in the images when the AKO level increased to 15–30%.

CLSM was employed to elucidate the microstructure of the HIPEs. As shown in Figure 2B, the oil droplets (red color) in the HIPE of soybean oil were spherical in shape and had a wide particle size distribution. The green fluorescence (casein particles) surrounded the surface of oil droplets and formed a dense and compact layer, indicating that the HIPE was an O/W emulsion. With the subsequent partial replacement of soybean oil by AKO, the oil droplets became smaller and more uniform, then began to deform, and eventually were densely packed. Replacing 5% of soybean oil by AKO caused an abrupt reduction in the size of oil droplets and a substantial improvement in uniformity of oil droplets. As the AKO level increased, the droplet size further decreased but tended to be stable and the oil droplets were closely packed at 10–20% levels of AKO. Liu et al. [7] reported that a smaller droplet size of HIPEs led to a closer crosslinking between oil droplets and a stronger emulsion gel that resisted its deformation and collapse. This agreed with what was observed in the present study. At the same time, the green fluorescence turned lighter and existed not only on the surface of oil droplets, but also in the aqueous phase, indicating PLs expelling some casein particles on the droplet surface to the aqueous phase. This was consistent with a previous finding that proteins preferentially adsorbed to oil–water interfaces at low surfactant levels due to their much higher adsorption energy per molecule, but at higher levels surfactants, preferentially adsorbed because of their higher packing efficiency [37].

The cryo-SEM images of HIPEs were analyzed to obtain further microstructure information of oil droplets in the HIPEs (Figure 2C). The figure showed that oil droplets in the HIPE of soybean oil were polydisperse spheres with a smooth surface. Casein particles formed a relatively dense film on the droplet surface, but crosslinking between droplets was not observed. With AKO replacing part of soybean oil, the droplets deformed and their size gradually decreased and became more uniform. When the AKO level was 15% and above, it was clearly seen that casein particles in the continuous phase bridged neighboring oil droplets and formed a three-dimensional crosslinked network structure in the emulsion that strengthened the emulsion gel properties [38] and impeded the collision and aggregation between the oil droplets, making the emulsion stable.

Figure 2D shows that the volume mean size of oil droplets in the HIPE of soybean oil was 16.07 ± 0.06 μm. When 5% of soybean oil was replaced by AKO, it abruptly decreased to 3.56 ± 0.01 μm and kept decreasing as the AKO level increased to 15 and 20%. However, a continual increase in the AKO level caused an increase in mean droplet size, which reached 10.47 ± 0.64 μm at the 30% AKO level. The trend of the change in mean droplet size could be explained by two opposite effects. On one hand, the PL content in the oil phase increased with increasing the AKO level. Because PLs are surfactants with a hydrophilic head and hydrophobic tail [39], which served as a co-emulsifier with casein to change the oil–water interface state, this effect tended to lower the droplet size. A previous study found that co-emulsifiers of sodium caseinate and lecithin could significantly reduce the particle size of thyme oil nanoemulsions [25]. Another study demonstrated that the addition of 0.5% (*w/w*) phosphatidylcholine to the water phase containing casein particles could reduce the Z-average diameter of fish oil-in-water emulsions from 330.3 ± 0.3 μm to 265.9 ± 0.4 μm (*p* < 0.05) [40]. On the other hand, because AKO has an extremely high viscosity, an increase in the AKO level increased viscosity of the mixed oil (Table 2), which tended to increase the mean droplet size, because a high viscosity prevents the effective mixing of the system and reduces the amount of dispersed phase that can be incorporated into the emulsion [41]. Due to these two opposite effects of increasing the AKO level on the mean droplet size, there existed an AKO level, which was around 15–20% under the present experimental conditions, that resulted in a minimum mean droplet size.

### 3.2. Effects of the AKO Level on Rheological Properties of the HIPEs

An understanding of emulsion rheology is essential for designing efficient food processing operations and emulsion-based foods with the desired physicochemical, sensory, and nutritional attributes, such as appearance, texture, and shelf life. Figure 3 depicts the storage modulus (G′) and loss modulus (G″) of the HIPEs with different levels of AKO in the oil phase. As shown in Figure 3A, G′ was always higher than G″ for all HIPEs in the linear viscoelastic regime in which both moduli were independent on the applied strain or stress, indicating clearly a gel-like behavior. This indirectly implied that densely packed oil droplets in the HIPEs were crosslinked and formed a three-dimensional network structure that filled the entire volume of the system, thereby leading to somewhat solid-like behavior [42]. This phenomenon was also observed in the independent frequency sweep tests depicted in Figure 3B.

Interestingly, replacing 5% of soybean oil by AKO substantially decreased G′, indicating a much higher fluidity of the mixed-oil HIPEs. However, a continual increase in the AKO level reversed the change in G′, and when the AKO level reached 15%, the HIPE of mixed oil had a higher G′ than the HIPE of soybean oil. This was consistent with changes in the size and surface appearance of oil droplets in the HIPEs as the AKO level increased (Figure 2). When 5% of soybean oil was replaced by AKO, although the oil droplets became less deformable (due to high viscosity of AKO) and some crosslinking occurred between them, the substantially increased smoothness of the droplet surface (due to loss of casein particles) reduced surface friction between droplets [43], thereby improving the fluidity of the emulsion. For a similar reason, Damanik and Murkovicn [44] reported that increasing lecithin concentration from 0.5% to 0.75% lowered viscosity of emulsions stabilized by chitosan. As the AKO level kept increasing, greatly increased gel structure (G′) of HIPEs was observed due to the decreased deformability of oil droplets and large amounts of crosslinking between the droplets.

### 3.3. The T_2_ Relaxation Times of the HIPEs

Low-field nuclear magnetic resonance (LF-NMR) provides an effective method of evaluating the dynamics of water mobility and distribution [45]. Figure 4 illustrates the *T*_2_ relaxation spectra of the HIPEs with different levels of AKO at 25 °C. As shown in the figure, the spectrum of a HIPE with 0–10% AKO had two components. One had a relaxation time of 65.52–193.71 (*T*_22_), which corresponded to the structural (partially movable) water entrapped in the emulsion gel system, the other represented the bulk (free) water in the HIPE and had a longer relaxation time of 261.43–498.69 (*T*_23_). When the AKO level exceeded 10%, a minor population with a short relaxation time of 0.52–1.50 ms (*T*_21_) appeared, which represented the bound (immovable) water. The peak area represents the fraction of water associated with that range of relaxation time. The *T*_2_ relaxation times and peak areas of the HIPEs with different levels of AKO at 25 °C are listed in Table 3. According to the table, *A*_22_ and *A*_23_, the peak areas of *T*_22_ and *T*_23_, respectively, for the HIPEs with 0–15% AKO were comparable with each other, indicating that bulk water was largely present in the emulsion, which explained the observed fluidity of the HIPEs (Figure 2A). As the AKO level increased, values of *T*_22_ and *T*_23_ decreased, revealing the reduced mobility of water. Figure 3 shows that the *T*_22_ and *T*_23_ peaks merged into one for the HIPEs with 25% and 30% AKO, with an area accounting for more than 95% of the total peak area, suggesting a compact structure of emulsions. A similar phenomenon was reported in a previous study that showed that the boundary of *T*_22_ and *T*_23_ became indistinctive, and the two peaks tended to merge into one with a range of relaxation times representing a transition from a fluid emulsion to a solid-like gel [46].

### 3.4. The Physical Stability of the HIPEs

The stability and water holding capacity of the HIPEs were determined by centrifugation. Centrifugation excludes the excess water from an emulsion, causing upwards migration of the dispersed phase. The creaming index (CI) and appearance of HIPEs with different levels of AKO after centrifugation are depicted in Figure 5. According to the figure, the CI of the HIPE of soybean oil was 17.56%, reflecting that centrifugation removed most of the excess water (totally 25%) from the emulsion. With the AKO level rising from 0% to 15%, the CI decreased significantly (*p* < 0.05) from 17.56% to 4.80%. From top to bottom, the HIPEs displayed three layers that were creamed layer, transparent serum layer, and caseins layer without oil-off was observed. This illustrated a gradual decrease in free water in the emulsions, which agreed with the LF-NMR analysis (Section 3.3), as well as the gradual buildup of crosslinked network structure within the HIPEs. When the AKO level was 15% and above, the HIPEs showed excellent water-holding capacity due to their dense and strong network structure (Figure 2B,C) resisting deformation during centrifugation and thus holding water tightly through capillary forces [7].

The stability of the HIPEs was also tested during 30 d of storage at 4 °C. Figure 6 illustrates the visual observation of HIPEs with different levels of AKO at 4 °C for 7 d and 30 d of storage, and their cryo-SEM morphologies after 30 d of storage. Figure 2A shows that the fresh HIPEs had a homogeneous appearance and most of them were capable of self-supporting except at 5% AKO. After 7 d of storage, Figure 6A shows that the HIPEs at 10% or lower AKO could not self-support but all others stayed stable. Even after an extended storage time (30 d), the HIPEs with 15–20% AKO still remained uniform although they lost more or less their self-supporting properties. As shown in the cryo-SEM images after 30 d of storage (Figure 6C), the droplet size of all HIPEs became larger compared with their fresh counterparts (Figure 2C). Especially for the HIPEs with 0–10% AKO, the microcosmic morphology of the droplets changed, large droplets appeared, and the casein particles protruded into the aqueous phase.

### 3.5. The Oxidative Stability of the HIPEs

The oxidative stability of the HIPEs was evaluated by using the Rancimat. During oxidation, the produced formic acid is transferred with the air stream to the measurement vessel filled with water and detected by an increase of the conductivity. The conductivity curves increased from moderate to sharp, and the intersection of the curves’ tangents is the oxidation induction time [47]. The longer the oxidation induction time, the better the oxidative stability of the samples [48]. Figure 7 depicts the oxidation induction time of HIPEs with different levels of AKO. As illustrated in the figure, increasing the AKO level from 5% to 30% did not lead to reduced oxidation induction time of HIPEs, although the contents of DHA and EPA, which are very susceptible to oxidation [49], in the mixed-oil phase substantially increased from 8.73 ± 0.35 and 5.79 ± 0.32 mg/g HIPE to 52.3 ± 2.08 and 34.73 ± 1.92 mg/g HIPE, respectively (Table 2), indicating that the gel-like structure of the HIPEs could delay oxidation of the unsaturated fatty acids in the emulsion matrix. This might be because the increased concentration of PLs in the mixed-oil phase resulted in a thicker oil–water interfacial layer, as indicated by Shao and Tang [50], who reported that a thicker viscoelastic interfacial membrane could retard lipid oxidation in emulsions. In addition, the lower molecular motion and energy transfer rate in a stronger gel-like structure of the HIPEs could also slow down lipid oxidation [31]. Nonetheless, due to the two opposite effects of increasing the AKO level on the oxidative stability of the HIPES, the HIPE with 25% AKO displayed the highest stability among all the samples examined.

## 4. Conclusions

With casein as a co-emulsifier, HIPEs of mixed soybean oil and AKO that accounted for up to 30% (*w/w*) of the oil phase could be prepared. The microstructure and properties of the HIPEs were affected by the AKO level. As the AKO level increased, the change in the mean oil droplet size followed a concave-up curve that displayed a minimum mean size at around 15–20% AKO, and the oil droplets became more densely packed. At the same time, CLSM micrographs showed that an increased portion of casein particles were expelled by PLs from the surface of oil droplets to the aqueous phase. As illustrated in both CLSM and cryo-SEM micrographs, the resultant densely packed small oil droplets were favorably bridged by the dispersed casein particles, resulting in a three-dimensional network structure of the HIPEs, which was also signified by the reduced *T*_2_ relaxation time, the somewhat solid-like behavior (G′ > G″), and the increased physical stability. The rheological tests demonstrated that at low levels (5 and 10%) AKO increased the fluidity of the HIPEs due to the increased surface smoothness of oil droplets, while at higher levels it did the opposite due to the high crosslink density between oil droplets. Lastly, increasing the AKO level decelerated the oxidation of the HIPEs. These findings provide useful insights for developing HIPEs of highly viscous AKO using a carrier oil (e.g., soybean oil) and a co-emulsifier (e.g., casein) and thus help expand applications of AKO in foods. The present study did not examine effects of temperature and pH on the stability of the HIPEs and packing of the emulsion droplets, which warrant further investigation.

## Figures and Tables

**Figure 1 foods-10-00917-f001:**
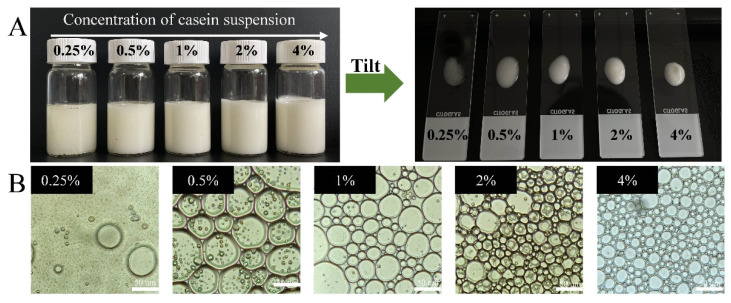
Appearance (**A**) and optical micrographs (**B**) of high internal phase emulsions (HIPEs) (φ = 0.75) of soybean oil under different casein concentrations in the continuous (water) phase (0.25, 0.5, 1, 2, and 4%, *w/w*). Scale bars, 50 µm.

**Figure 2 foods-10-00917-f002:**
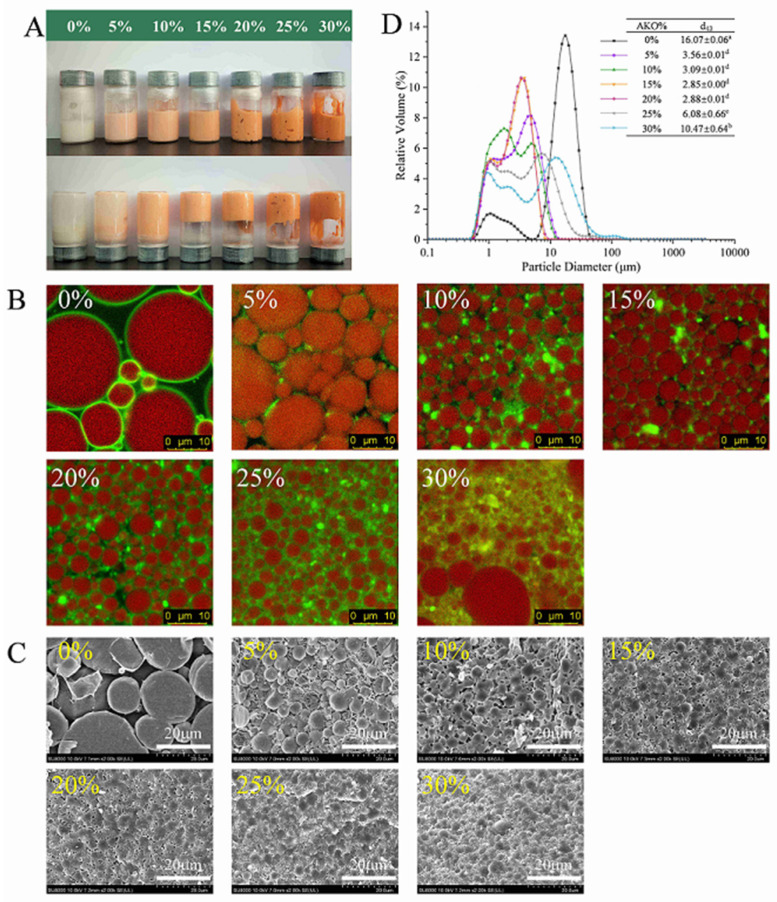
Appearance (**A**), CLSM observation (Scales bars, 10 μm) (**B**), cryo-SEM observation (Scales bars, 20 μm) (**C**), and particle size distributions (**D**) of HIPEs with different levels of Antarctic krill oil (AKO) (0, 5, 10, 15, 20, 25, and 30%, *w/w*). In the CLSM micrographs, oil droplets were stained with Nile red and shown in red at 488 nm, and casein particles were stained with Nile blue A and shown in green at 633 nm. Scale bars, 10 µm. Values with different letters (a–d) are significantly different (*p* < 0.05).

**Figure 3 foods-10-00917-f003:**
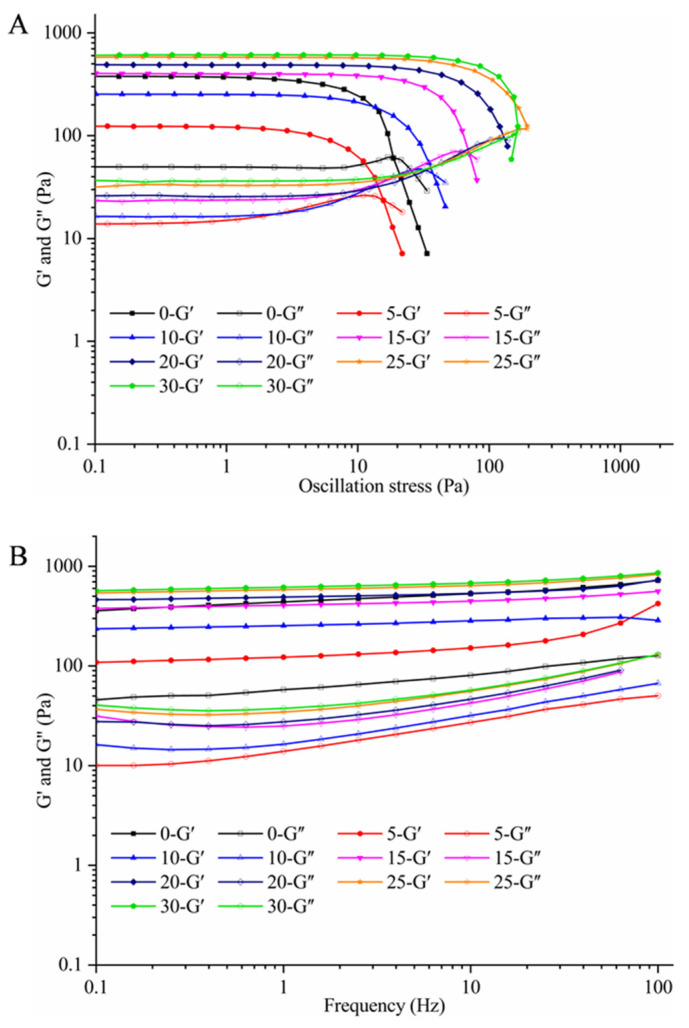
The storage modulus (G′, Pa) and loss modulus (G″, Pa) as a function of oscillation stress (Pa) with a constant frequency between 0.1 and 10 Hz (**A**) and as a function of frequency (Hz) with a constant strain of 1.00% (**B**) for HIPEs with different levels of Antarctic krill oil (AKO) (0, 5, 10, 15, 20, 25, and 30%, *w/w*).

**Figure 4 foods-10-00917-f004:**
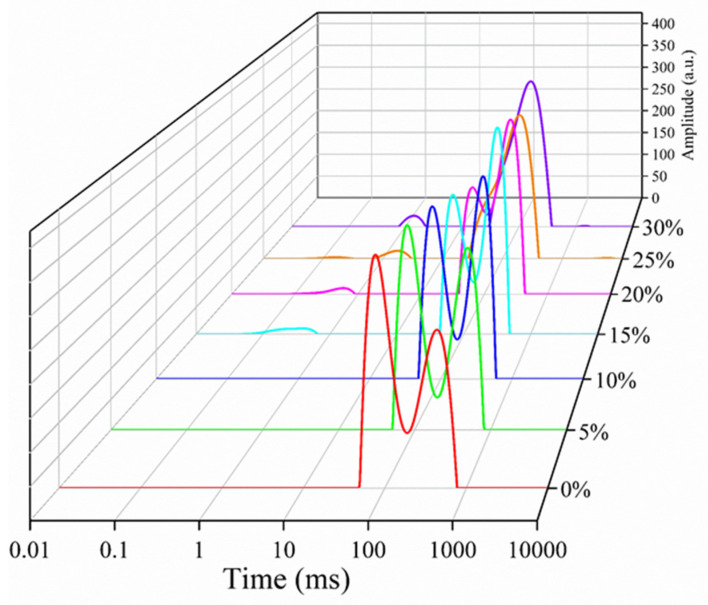
The distributions of *T*_2_ relaxation times of the HIPEs with different levels of Antarctic krill oil (AKO) (0, 5, 10, 15, 20, 25, and 30%, *w/w*) at 25 °C.

**Figure 5 foods-10-00917-f005:**
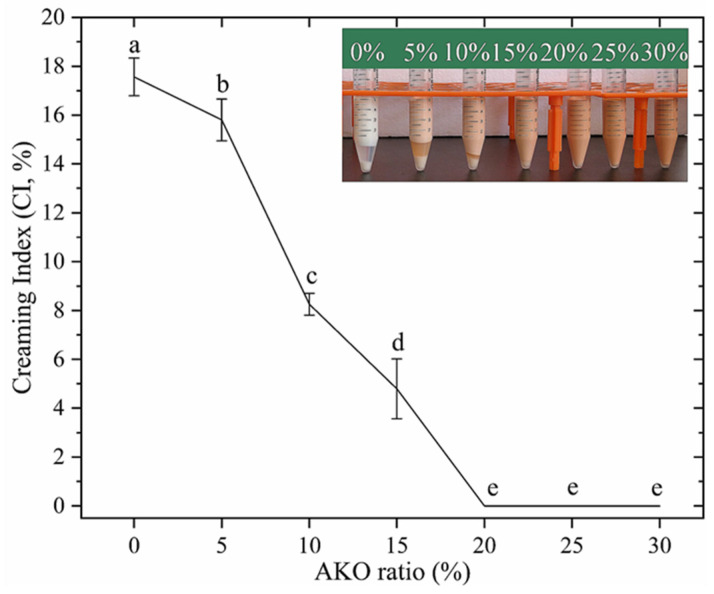
The creaming index (CI) and appearance (insert, after centrifugation) of HIPEs with different levels of Antarctic krill oil (AKO) (0, 5, 10, 15, 20, 25, and 30%, *w/w*). Values with different letters (a–e) are significantly different (*p* < 0.05).

**Figure 6 foods-10-00917-f006:**
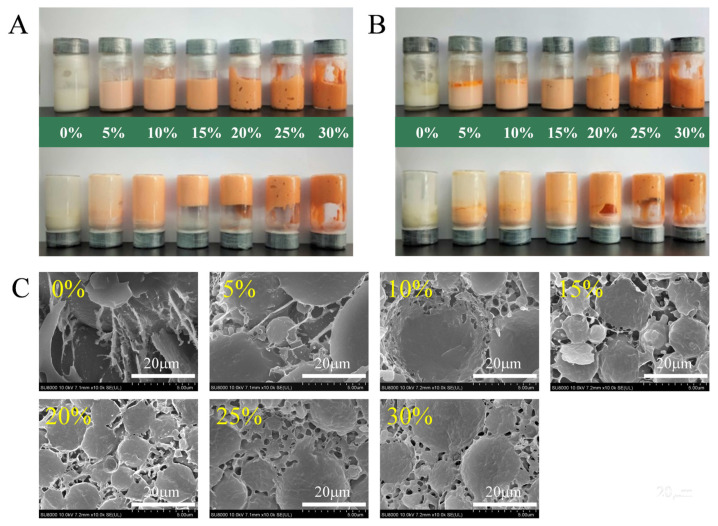
Visual observation of the HIPEs with different levels of Antarctic krill oil (AKO) (0, 5, 10, 15, 20, 25, and 30%, *w/w*) at 4 °C for 7 d (**A**), 30 d (**B**), and the cryo-SEM morphology of the HIPEs after 30 d of storage (**C**). Scale bars, 5 μm.

**Figure 7 foods-10-00917-f007:**
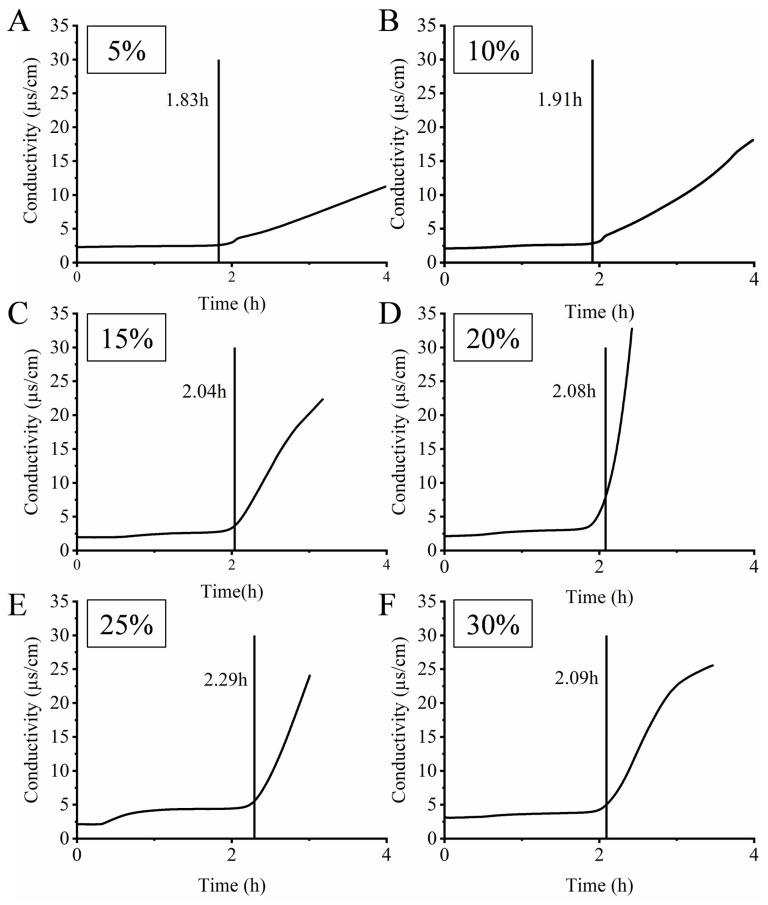
Oxidation induction time of HIPEs (**A**–**F**) with different levels of Antarctic krill oil (AKO) (5, 10, 15, 20, 25, and 30%, *w/w*) subjected to accelerated oxidation at 120 °C under a 20 L/h air flow.

**Table 1 foods-10-00917-t001:** Preparation of HIPEs using a casein suspension (2, 1, 0.5, and 0.25% casein, *w/w*) and soybean oil or mixed oil (soybean oil + AKO).

HIPE Sample	Component	Oil Phase (%, *w/w*)	
		Soybean Oil	AKO
0% HIPE	Casein suspension (10 mL) + Oil (30 mL)	100	0
5% HIPE		95	5
10% HIPE		90	10
15% HIPE		85	15
20% HIPE		80	20
25% HIPE		75	25
30% HIPE		70	30

**Table 2 foods-10-00917-t002:** The phospholipids (PLs) content, viscosity, and DHA and EPA content in the oil phase containing different levels of Antarctic krill oil (AKO). In each column, different letters denote significant difference (*p* < 0.05).

AKO Levels in the oil Phase (%, *w/w*)	PLs Content in the Oil Phase (mg/g)	Viscosity of the oil Phase (mPa·s)	DHA Content in the Oil Phase (mg/g)	EPA Content in the Oil Phase (mg/g)
0	0.03 ± 0.00 ^a^	55.52 ± 1.34 ^a^	N.D.	N.D.
5	21.27 ± 0.05 ^b^	59.34 ± 0.41 ^a^	8.73 ± 0.35 ^a^	5.79 ± 0.32 ^a^
10	41.45 ± 0.05 ^c^	62.84 ± 1.71 ^a^	17.46 ± 0.69 ^b^	11.58 ± 0.64 ^b^
15	70.66 ± 0.21 ^d^	69.93 ± 0.41 ^a^	26.19 ± 1.04 ^c^	17.37 ± 0.96 ^c^
20	85.52 ± 0.27 ^e^	77.05 ± 1.28 ^a^	34.92 ± 1.39 ^d^	23.15 ± 1.28 ^d^
25	123.57 ± 0.00 ^f^	84.20 ± 0.41 ^a^	43.65 ± 1.73 ^e^	28.94 ± 1.60 ^d^
30	154.12 ± 0.32 ^g^	91.01 ± 1.61 ^a^	52.38 ± 2.08 ^f^	34.73 ± 1.92 ^e^
100	507.05 ± 0.00 ^h^	2025.00 ± 59.23 ^b^	174.62 ± 6.93 ^g^	115.77 ± 6.39 ^f^

N.D. Not detectable.

**Table 3 foods-10-00917-t003:** The *T*_2_ relaxation time and peak area of HIPEs with the oil phase (soybean oil) containing different levels of Antarctic krill oil (AKO) at 25 °C. In each column, different letters denote significant difference (*p* < 0.05).

AKO Levels (%, *w/w*)	*T*_21_ (ms)	*T*_22_ (ms)	*T*_23_ (ms)	*A* _21_	*A* _22_	*A* _23_
0	N.D.	77.00 ± 0.00 ^c^	434.24 ± 0.00 ^d^	N.D.	12,879.86 ± 45.09 ^e^	9890.16 ± 3.64 ^a^
5	N.D.	80.63 ± 0.00 ^d^	498.69 ± 0.00 ^e^	N.D.	11,725.34 ± 12.35 ^d^	11,138.55 ± 4.72 ^b^
10	N.D.	77.00 ± 0.00 ^c^	395.97 ± 0.00 ^c^	N.D.	10,297.69 ± 19.09 ^c^	11,774.26 ± 1.55 ^c^
15	0.52 ± 0.02 ^a^	68.61 ± 0.00 ^b^	319.30 ± 4.24 ^b^	391.65 ± 66.10 ^a^	8886.73 ± 33.99 ^b^	12,932.71 ± 11.36 ^e^
20	0.61 ± 0.02 ^a^	65.52 ± 0.00 ^a^	261.43 ± 0.00 ^a^	548.06 ± 10.71 ^b^	7538.46 ± 126.66 ^a^	14,124.78 ± 86.55 ^f^
25	1.65 ± 0.09 ^c^	193.71 ± 0.00 ^g^	N.D.	639.77 ± 16.04 ^c^	19,888.21 ± 25.66 ^f^	N.D.
30	1.50 ± 0.05 ^b^	161.07 ± 0.00 ^f^	N.D.	720.92 ± 17.70 ^d^	20,821.39 ± 78.61 ^g^	N.D.

N.D. Not detectable.

## Data Availability

Not applicable.

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
