# Peer review of "Development of a High Internal Phase Emulsion of Antarctic Krill Oil Diluted by Soybean Oil Using Casein as a Co-Emulsifier"

_foods, 2021, doi:10.3390/foods10050917_

Round 1

Reviewer 1 Report

The manuscript is well written and contains enough interesting data. While the design of the experiments and the choice of variables seems to me to be correct, I consider that some of the measurements and experiments could have been made more profitable.

  • Abstract and introduction are conceasive and provide information on the purpose of work and background within the line of research.
  • 2.2. Why have you used these sample preparation conditions? Is it based on previous work? Perhaps a stage of optimization of processing in homogenization speed or time would allow for better results.
  • 2.4. Why has no Sauter diameter or span or uniformity data been included?
  • 2.5. Since frequency sweeps have been performed, why have no flow curves been made?
  • Figure 2 looks like a "data puzzle". In addition, Figure 2D could provide much more information (standard deviations, span, sauter...) and also does not correspond to droplet size distributions, which should be provided. I suggest removing figure 2E and adding a new figure with the true droplet size distributions, medium diameters with standard deviations, span...
  • The rest of the results and discussion, as well as the conclusions, are consistent and well written.

Author Response

Dear reviewer, please check the attached  file. Thank you very much.

Reviewer 2 Report

I believe this manuscript explores an interesting topic with good impacts to the field. Nonetheless, I believe the manuscript can have a better exposure to the food community by at least addressing some of the major comments illustrated as follows: 

(1) The authors need to add a discussion on why they are using soybean oil as a diluent. To clarify my point, why not using other types of oils? 

(2) The introduction is missing the literature reports on the effects of experimental conditions such as emulsification process, pH, and temperature on the stability and rheological properties of HIPE.  

(3) The authors show that an increase in AKO content resulted in reducing the emulsion droplet size and then at certain point it had an opposite effect; like U-shape? I am not very convinced that the reason for increase in the droplet size is due to the mixing efficiency. If this is really the case, the higher shearing conditions should fix the problem (this can be explored). The authors need to propose other possible scenarios as well. For example, could this be due to the presence of impurity in AKO or even enhancing the droplets packing which might increase the chance of collision? 

(4) Did the authors use Nile Blue A to label casein particles? It seems a bit confusing. Please clarify this in the manuscript. 

(5) How do the temperature and pH affect the stability and packing of the emulsion droplets (crosslinking)? At least something to consider for the future work and should be mentioned this in the paper for future directions. 

(6) The authors need to highlight the importance of this work in a more systematic fashion and emphasize how this is different than other studies.

(7) The scale bars cannot be seen easily on the figures. Please improve the quality of the scale bars. Additionally, the oxidation induction time results shown in Figure 7 need to be labeled properly (e.g., A=5% AKO, B=10% AKO, etc.). 

(8) The manuscript requires proofreading, there are several grammatical as well as typographical issues in the text. 

(9) Consider citing few articles published in this journal, Foods. 

(10) The literature review can be improved. The following work can be considered as such:

(i) "Oat protein solubility and emulsion properties improved by enzymatic deamidation." Journal of Cereal Science 64 (2015): 126-132.; (ii) "Effects of high pressure homogenization on faba bean protein aggregation in relation to solubility and interfacial properties." Food Hydrocolloids 83 (2018): 275-286.

Author Response

Dear reviewer,please check the attached file. Thank you very much.

Reviewer 3 Report

The Author proposed how to develop selected high internal phase emulsion. The idea of the experiments is interesting, but the description and presentation of the results must be strongly improved. First of all, state do you consider the emulsion system as a Pickering type or not. If yes, define what will be solid particles that will stabilize the emulsion. Then, consider providing additional experiments of the stabilizer characterization. More detailed comments are listed below. 

Explain all short names when used for the first time i.e. CLSM and cryo-SEM.

Introduction

  1. 31 - this sentence is not clear please specify what will be delivered. In NExt sentences, please explain what you mean by the delivery and present more clearly what you talking about (I can t see the connection whit delivery systems).
  2. 37 - 0,74 of what?
  3. 39 - what kind of bioactive?
  4. 46 - please add what kinds of products you mean

Rewrite the last part of the introduction. Furthermore, cross out the last sentence "To the best of our knowledge, there have been no reports of using HIPEs to carry AKO." - it is not necessary.

Section 2.2.

  • it is almost impossible to verify your experiment - please write in detail how the samples were performed (including concentrations of the compounds - add a table with sample description).
  • l. 100 give details about the light microscope
  • l. 150 improve the equation

Results and discussion

  • what does it mean a-d in fig 2 referring to particle size?
  • improve the quality of cryosem images and add the scale bars
  • change the numbering in fog 2. particle size should be B
  • particle size distribution plots should be added - z-ave doesn't inform about the particle size distribution

Please clearly state do you think your emulsion will be Pickering? If yes, please add more comments. What kind of solid particles you consider as an emulsion stabilizer and please add comments on the microscopic images where are they (mark them) I could not recognize it on the presented images.

Author Response

(The authors gave the same response as above.)

Round 2

Reviewer 1 Report

I consider that, in its current state, this manuscript can be considered for publication on Foods. Thank you to the authors for their comments and for responding to suggestions.

Author Response

Thank you very much. We feel lucky that our manuscript went to you as the valuable comments from you helped us with the improvement of our manuscript.

Reviewer 2 Report

The authors addressed my comments to a good extent. So, I believe the manuscript is suitable for publication after final round of proofreading. 

Author Response

(The authors gave the same response as above.)

Reviewer 3 Report

The Authors improved the manuscript. Before publication, some minor changes are recommended. Detailed comments are listed below:

  • please add why "AKO is very viscous, it was diluted by soybean oil" why soybean oil?
  • still, I don t like the sentence "we first, etc.", but this is not the case of this manuscript
  • please add Table S1. to the main text (as I good remember there is no signs and tables limit in MDPI Journals)

Author Response

Thank you very much for your valuable comments to improve our manuscript. Please check the attached file.
